# Characterising lithium-ion electrolytes via operando Raman microspectroscopy

Jack Fawdon[1], Johannes Ihli[1,2], Fabio La Mantia [3] & Mauro Pasta [1,4✉]

Knowledge of electrolyte transport and thermodynamic properties in Li-ion and beyond Li-ion technologies is vital for their continued development and success. Here, we present a method for fully characterising electrolyte systems. By measuring the electrolyte concentration gradient over time via operando Raman microspectroscopy, in tandem with potentiostatic electrochemical impedance spectroscopy, the Fickian "apparent" diffusion coefficient, transference number, thermodynamic factor, ionic conductivity and resistance of charge-transfer were quantified within a single experimental setup. Using lithium bis(fluorosulfonyl) imide (LiFSI) in tetraglyme (G4) as a model system, our study provides a visualisation of the electrolyte concentration gradient; a method for determining key electrolyte properties, and a necessary technique for correlating bulk intermolecular electrolyte structure with the described transport and thermodynamic properties.

[1] Department of Materials, University of Oxford, Oxford, United Kingdom. [2] Paul Scherrer Institut, Villigen, Switzerland. [3] Universtät Bremen, Energiespeicher-und Energiewandlersysteme, Bremen, Germany. [4] The Faraday Institution, Quad One, Harwell Science and Innovation Campus, Didcot, UK . ✉email: mauro.pasta@materials.ox.ac.uk

Since the commercial development of lithium-ion batteries (LIBs) in the early 1990s, they have been employed in many applications that society now depends on.[1] Research into "beyond lithium-ion" technologies, such as lithium metal batteries (LMBs), has also surged in popularity in recent years, with demand for secondary battery technologies increasing considerably as governments aim to become carbon neutral.[2] Battery performance is highly dependent on the transport and thermodynamic properties of their electrolyte.[3] In LMBs specifically, the uniformity of lithium plating is severely affected by the transport properties of the electrolyte, thus playing a crucial role in the formation and propagation of lithium dendrites.[4,5] It is therefore of paramount importance to be able to accurately measure and ultimately improve these properties for the continued development of innovative battery technologies.

For a full understanding of an electrolyte system, the ionic conductivity ($\kappa$), Fickian "apparent" diffusion coefficient ($D_{app}$), cationic transference number ($t_+^0$), molar thermodynamic factor ($\chi_M$) and interfacial charge-transfer resistance ($R_{ct}$) are key parameters.[6–8] Each of these concentration-dependent properties provides information toward a comprehensive picture of the electrolyte's transport, thermodynamic state, and interfacial kinetics.

$D_{app}$ and $t_+^0$ dictate whether, and to what extent, concentration gradients across cells form. The formation of such gradients leads to concentration overpotential ($\eta_c$), which can be harmful to all battery types' performance, especially their rate capability.[9,10] $D_{app}$ is characterised through Fick's laws of diffusion, and $t_+^0$ is defined as the proportion of current carried by the cation. Each is conventionally measured through different electrochemical techniques, with $D_{app}$ occasionally assessed using restricted-diffusion cells,[7,11,12] and $t_+^0$ from either the Hittorf method[7,11,13–15], pulsed-field gradient NMR[16] or a "steady-state current" method, like the Bruce–Vincent.[17–19] Each have their limitations, with Hittorf often requiring large volumes of electrolyte and not being able to measure in situ concentration changes, and steady-state methods making assumptions such as the electrolyte being infinitely dilute (perfectly ideal).[20]

$\chi_M$ provides a link between observed concentration and thermodynamic activity.[21,22] It is influenced by ion association, with the formation of solvent-separated ion pairs (SSIPs), contact ion pairs (CIPs) and aggregates (AGGs), and the extent and orientation of solvation. It is classically explored by measuring the liquid-junction potential in a concentration cell, between a "test" concentration and a "reference" concentration[7,13–15,23]. The measurement is quite rarely performed when characterising new electrolytes but is of critical importance for understanding how its thermodynamic state is influenced by electrolyte structure.

In addition, with electrolyte intermolecular structure influencing interfacial resistances, through charge-transfer, and compositional influence on the solid electrolyte interphase (SEI),[24] measurings $R_{ct}$ is critical.[8,25]

While others have reported individual setups for measuring electrolyte properties, here we disclose a unified method for determining $\kappa$, $D_{app}$, $t_+^0$, $\chi_M$ and $R_{ct}$. This was achieved by visualising and fitting electrolyte concentration gradients using operando scanning Raman microspectroscopy, in tandem with potentiostatic electrochemical impedance spectroscopy (PEIS). The fitting of concentration gradients over time allowed for the direct measurement of the evolving diffusion length ($L_d$) and interfacial gradient ($dc_s/dz|_{z = 0,L}$), which led to the determination of $D_{app}$ and $t_+^0$ respectively. Also, by monitoring the progressing concentration at each cell extreme as the gradient was forming, and using chronopotentiometry (CP) data and PEIS to calculate $\eta_c$, $\chi_M$ was calculated. This is the first time $\chi_M$ has been measured in such a dynamic setup. Moreover, by measuring $D_{app}$, $t_+^0$, $\kappa$ and the $\chi_M$, a detailed picture of the multi-component electrolyte system was obtained through Stefan–Maxwell coefficients,[21] which describes the frictional interaction between species. Our described setup is advantageous for numerous reasons: namely, the amount of information one can obtain in a single experiment; the small instrumental error involved by having one method to measure all properties; the small volume of electrolyte required (<1 mL), reducing waste and allowing for analysis even at small scale, and its compatibility with concentrated-solution theory.[21]

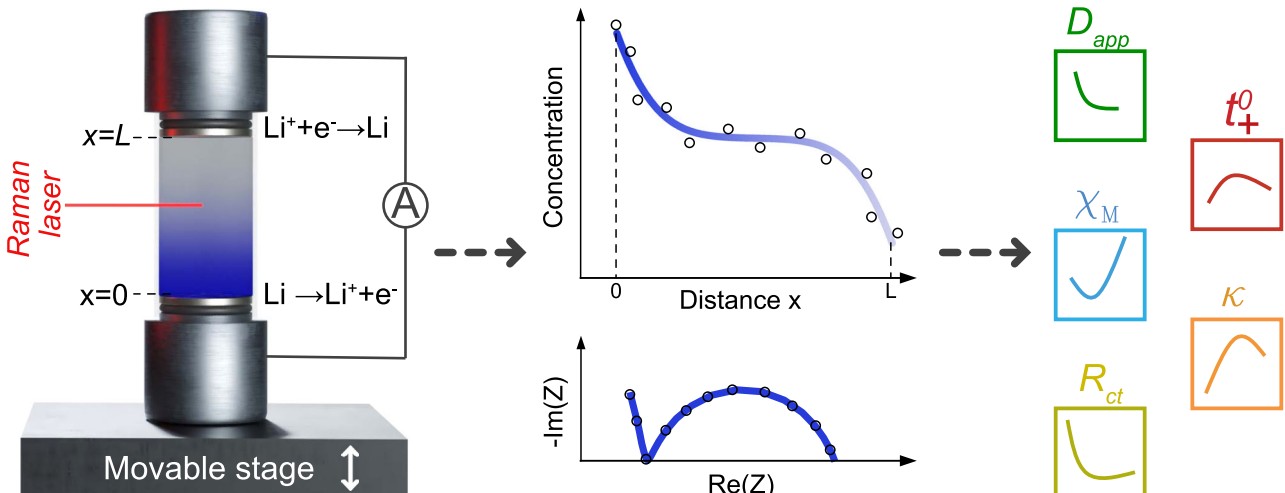

**Fig. 1 Schematic of the method to measure key electrolyte properties.** Li$^+$ concentration ($c_s$) gradient formation in an optical Li/Li symmetric cell, while current is passed. Using a scanning confocal Raman microspectrometer, a time-series of one-dimensional line scans was performed to obtain a progression of concentration gradients; measured with the stage moving in the $z$-direction. The cell was placed vertically on the stage to avoid natural convection. Potentiostatic electrochemical impedance spectroscopy (PEIS) was also performed prior to passing current to quantify ionic conductivity ($\kappa$) and resistance of charge-transfer ($R_{ct}$), and between line scans to obtain an accurate concentration overpotential ($\eta_c$). Using the diffusion length ($L_d$), interfacial gradient ($dc_s/dz|_{z = 0,L}$) and $\eta_c$ from each concentration profile, the Fickian "apparent" diffusion coefficient ($D_{app}$), cation transference number ($t_+^0$), and molar thermodynamic factor ($\chi_M$), were calculated.

Importantly, by determining these electrolyte properties using a spectroscopic technique such as Raman microspectroscopy, microscopic electrolyte structural information can be correlated to macroscopically determined phenomena such as those listed above. Although there have been some Raman spectroscopy studies measuring [Li+],[26–29] this is the first that measures the entire concentration gradient formation as current is passed, with the extraction of critical electrolyte properties. X-ray[30] and NMR[31–33] techniques have also been recently used to measure concentration gradients, with an aim of understanding $t_+^0$ specifically. These methods are often unavailable to many researchers, so our technique provides a more readily available alternative. Furthermore, our focus here is to fully characterise the electrolyte by clearly measuring five key parameters, when other studies only calculate one or two.

We applied our method to lithium bis(fluorosulfonyl)imide (LiFSI) in tetraethylene glycol dimethyl ether (tetraglyme—G4) over a concentration range of 0.25–2 m. LiFSI is an increasingly popular salt used in lithium metal anode studies due to its low viscosity in typical solvents and desirable decomposition products (e.g., LiF) that form the SEI.[8] Fluoride-rich SEI layers have shown to be effective at suppressing dendrite growth due to the electronic insulation and high surface energy of LiF.[34,35] Glymes are also popular for use against lithium metal (in LMBs) because of their cathodic stability and low viscosity. G4 has been shown to exhibit low flammability, low volatility and cathodic stabilities below 0 V vs. Li+/Li.[36] They have been studied for their use in Li–O2 and Li–S batteries in particular.

In summary, this study demonstrates a novel experimental setup that uses operando Raman microspectroscopy to measure concentration gradients, enabling the determination of fundamental concentration-dependent electrolyte properties: $\kappa$, $R_{ct}$, $D_{app}$, $t_+^0$ and $\chi_M$. Using LiFSI in G4, each described electrolyte property is measured and reported, gaining valuable insight into the electrolyte system. We hope our study provides a foundation for the full characterisation of electrolytes using operando Raman microspectroscopy.

## Results

The developing Li+ concentration gradient between the electrodes of a symmetric lithium cell was determined using a scanning confocal Raman microspectrometer (Renishaw inVia Reflex); specifically acquiring a time-series of one-dimensional (1D)

Raman lines scans across the interelectrode space of a custom-built optical cell, with 8 mm diameter electrodes, while the current was passed. Line scans, consisting of 50 scanning points covering a distance of 1.5 cm, were performed every 4 h for 24–48 h depending on the LiFSI concentration being investigated. Concentrated electrolytes required a longer investigation period for a substantial gradient to form. Each line scan took 20 min to complete. Measurements were performed at 20 °C.

Through isolation of a concentration sensitive FSI− Raman band, we were able to equate each Raman spectrum in a line scan to the local Li+ concentration. This was following an instrument calibration of LiFSI solutions of known concentration, equating [FSI−] with [Li+]. Specifically, we normalised each spectral peak against the 1471 cm−1 –CH2 bending/scissoring mode solvent peak height, and correlated the 717 cm−1 FSI− S–N–S bend peak area with concentration. The increasing area of the 717 cm−1 peaks is illustrated in Fig. 2b, along with the emergence of CIPs and AGGs with increasing concentration, as the shoulder peaks begin to dominate. The calibration curve is illustrated in Fig. 2c. Also labelled in Fig. 2a is the peak at 868 cm−1; as noted in Supplementary Fig. 8a, its increasing intensity with concentration signifies the complexation of Li+ ions with G4, [Li(G4)]+. The 868 cm−1 peak represents the breathing mode of this crown ether-like structure.[36] The description of the intermolecular structure that these peaks provide will be used in subsequent sections to understand each macroscopic trend.

By fitting the acquired spectra and the resulting concentration gradients we were able to extract $D_{app}$, $t_+^0$ and $\chi_M$ of the studied electrolyte system. Complementary PEIS measurements at open circuit potential, first after an initial rest period of 4 h, and then interspaced between Raman line scans, enabled us to measure $\kappa$ and the evolution of $R_{ct}$. The latter allowing an accurate measurement of the concentration overpotential ($\eta_c$) over time.

Figure 1 illustrates both the setup used and the extracted properties that were measured. Important to note, the operando cell was placed vertically on the scanning stage, with plating Li+ occurring at the top and stripping from the bottom of the cell. This avoided detrimental natural convection, as illustrated in Supplementary Fig. 3a. To ensure the validity of the diffusion equation boundary conditions (Eqs. (1)–(3)) used to extract the stated parameters, an interelectrode distance of 1.5 cm was chosen. The optimal measurement current density was determined to be 100 μA cm−2, ensuring the development of a concentration gradient detectable by the instrumentation, while equally avoiding

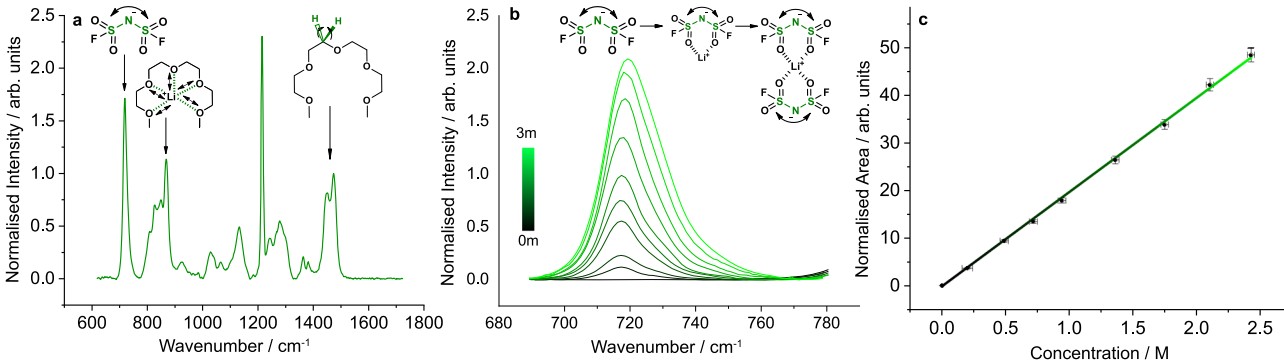

**Fig. 2 Intermolecular structure of lithium bis(fluorosulfonyl)imide (LiFSI) in tetraglyme (G4), and correlation of 717 cm−1 FSI− peak area with [Li+], as determined by Raman microspectroscopy. a** Raman spectrum of 2 m LiFSI in G4 normalised to peak intensity at 1471 cm−1; identifying the 717 cm−1 FSI− S–N–S bending mode peak, 868 cm−1 [Li(G4)]+ crown ether-like breathing mode peak, 1471 cm−1 –CH2 bending/scissoring mode peak. **b** Normalised 717 cm−1 peak, showing broadening of the peak with concentration, indicating the evolution of CIPs and AGGs, as illustrated. **c** Calibration curve equating normalised 717 cm−1 peak area to concentration, which was used to determine the concentration gradient via operando Raman microspectroscopy 1D line scans. Error bars represent the propagated standard error from the balance and density measurements (x-axis) and area calculation from three separate Raman acquisitions (y-axis).

dendrite formation. Instrumentation and acquisition details are provided in Supplementary Figs. 1–3 and the "Methods".

To validate and test the described methodology, a 1 m LiFSI in the G4 system was chosen as a model system. Molalities were used for increased reliability and accuracy of electrolyte preparation; for density measurements, and molarity equivalents see Supplementary Method 1.

Figure 3a shows the concentration gradient of the 1 m electrolyte after 12 h, illustrating a clear drop in concentration on the plating side and an equally noteworthy concentration increase on the stripping side. Profiles were fitted using non-linear least-squares minimisation with equation (1). Equation (1) is a solution to the diffusion equation,[21,37] in a symmetric cell setup, using the cation flux law as a spatial boundary condition. A full derivation of this equation is presented in Supplementary Note 1, Eqs. (6)–(19).

$$c_s(z,t) = c_s^* + a\left\{ \left(\frac{b}{\pi^{\frac{1}{2}}}\right) \exp\left(-\frac{z}{b}\right)^2 - z\,\mathrm{erfc}\left(\frac{z}{b}\right) - \left(\frac{b}{\pi^{\frac{1}{2}}}\right) \exp\left(-\frac{(-z+L)}{b}\right)^2 + (-z+L)\,\mathrm{erfc}\left(\frac{-z+L}{b}\right) \right\}$$

(1)

Where:

$$a\left(= \frac{dc_s}{dz}\bigg|_{z=0,L}\right) = \frac{J(1-t_+^0)}{nFD_{app}}\left(1 - \frac{\partial \ln c_0}{\partial \ln c_s}\right)^{-1}$$

(2)

$$b(= L_d) = 2(D_{app}t)^{\frac{1}{2}}$$

(3)

$c_s^*$ is the initial concentration, $c_s$ is the concentration of the salt, $t$ is time, $L$ is the distance between the electrodes, $J$ is the applied current density, $n$ is the number of moles, and the final factor in parentheses represents how solvent concentration is changing with respect to salt concentration at the interface, which we

termed the "solvent velocity factor". By including this factor, the cationic transference number is labelled $t_+^0$. If not referenced against the solvent velocity, $t_+$ is used. Supplementary Fig. 6b illustrates how this factor changes with concentration, showing that its inclusion becomes increasingly important at high concentrations. $b$ is equal to the diffusion length ($L_d$), and $a$ as the interfacial salt concentration gradient ($dc_s/dz|_{z=0,L}$). $c_0$ is equal to $(1 - V_s c_s)/V_0$, where $V_s$ and $V_0$ are the partial molar volumes of the salt and solvent respectively.[21] Supplementary Method 2 describes how $V_s$ and $V_0$ were calculated, and how the resulting solvent concentration is distributed across the cell (Supplementary Fig. 6c).

Figure 3b shows concentration profiles with an 8 h gap between each line scan. As predicted, the interfacial gradient $dc_s/dz|_{z=0,L}$ remained constant. Also one can note $L_d$ extending into the centre of the cell with increasing time.

To measure $D_{app}$, $L_d$ was plotted vs. $t^{1/2}$ in Fig. 3c, and the points were fitted to $L_d = (Dt)^{1/2}$, with the error bars representing standard error from Eq. (1) fitting. From the weighted gradient fitting, $D_{app}$ was calculated as $7.22 \pm 0.55 \times 10^{-11}$ m$^2$ s$^{-1}$. This compared well to the often-used method in the literature that follows the semilog open-circuit voltage (OCV) vs. time,[7,14] which we calculated as $6.98 \times 10^{-11}$ m$^2$ s$^{-1}$. See Supplementary Note 5 for further details.

$t_+^0$ was calculated from $dc_s/dz|_{z=0,L}$ (Eq. (2)), using $D_{app}$ from the previous calculation. Because $dc_s/dz|_{z=0,L}$ remained constant over the measured time frame, $t_+^0$ was quantified from each measured gradient and averaged along with its standard error. Due to the concentration differences on each side of the cell, the solvent velocity factor from equation (2) was slightly different at each interface, leading to somewhat different $t_+^0$ values throughout the cell. Nevertheless, due to the error from the fitting, this difference was deemed negligible for the calculation. The solvent velocity factor at the central point of the cell was therefore used, which was equal to 1.08 for 1 m. $t_+^0$ was subsequently

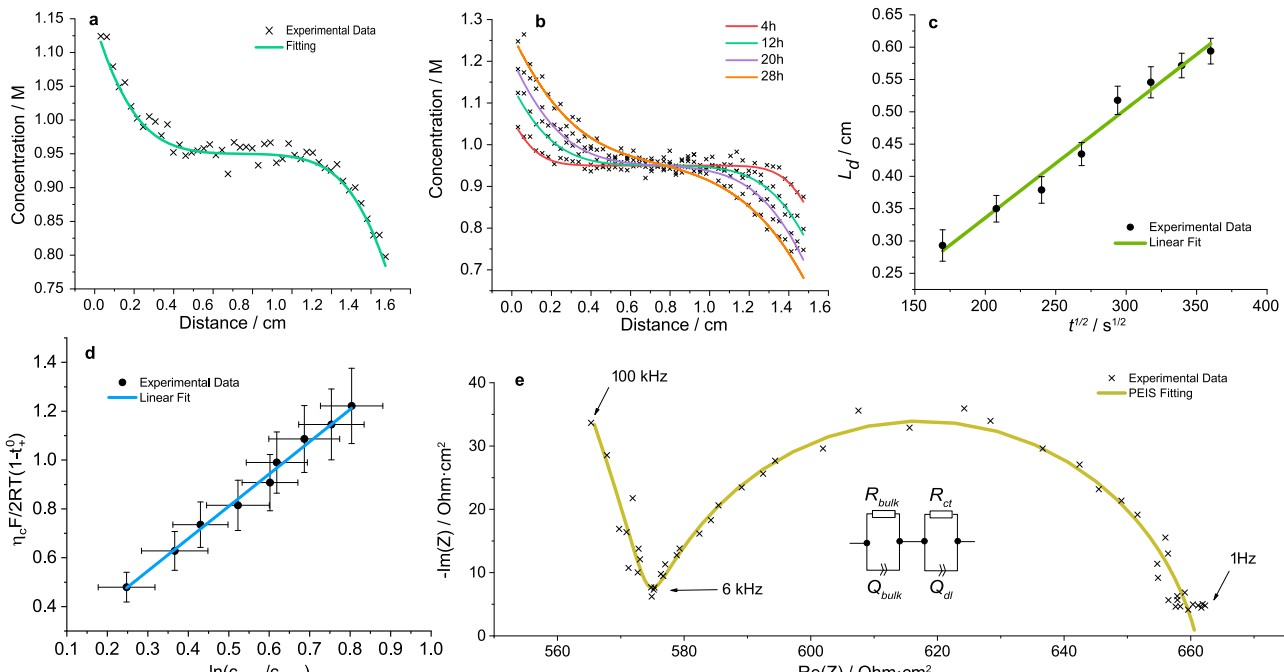

**Fig. 3 Model system 1 m LiFSI in G4, data analysis. a** The Li$^+$ concentration profile in 1 m LiFSI in G4 electrolyte after applying 100 μA cm$^{-2}$ of current for 12 h, with fitting (Eq. (1)). **b** Progression of a concentration gradient in 1 m electrolyte, after 4 h, 12 h, 20 h, 28 h. **c** Movement of Li$^+$ diffusion length ($L_d$) with time$^{1/2}$, showing linear behaviour for the $D_{app}$ calculation. $D_{app} = 7.22 \pm 0.55 \times 10^{-11}$ m$^2$ s$^{-1}$. **d** Plotting Eq. (4), illustrating how the $\eta_c$ function changes linearly with respect to the natural log of concentration ratio of each cell extreme, to measure $\chi_M$. $\chi_M = 1.41 \pm 0.31$. **e** PEIS Nyquist plot, with the equivalent circuit, of 1 m electrolyte, prior to current being passed, after 4 h rest, equating to $\kappa = 2.70 \pm 0.03$ mS cm$^{-1}$ and $R_{ct} = 92 \pm 12$ Ω cm$^2$.

calculated as $0.352 \pm 0.056$. The values compared well to a Hittorf-style analysis using the same setup, where $t_+^0$ was calculated as $0.392 \pm 0.006$. See Supplementary Note 6 for more details. To check the reliability of the method the measurement was repeated twice more.

From the fitted profiles, the concentration at each interface was measured and using Eq. (4) below[21] $\chi_M$ was calculated from the gradient of a weighted linear fit, see Fig. 3d. The error bars represent uncertainty in interfacial concentration from the fitted profiles in the x-axis and error in the $t_+^0$ calculation in the y-axis.

$$\chi_M = 1 + \frac{\ln f_\pm}{\ln c} = \frac{F}{2RT(1 - t_+^0)} \frac{d\eta_c}{d\ln \frac{c_{s,z=L}}{c_{s,z=0}}} \quad (4)$$

Where $f_\pm$ is the molar activity coefficient. Using the PEIS data prior to each line scan, $\eta_c$ was calculated by $\eta_c = \eta_{total} - I(R_{bulk} + R_{ct})$, where $\eta_{total}$ is measured from the chronopotentiometry data. Supplementary Fig. 8b illustrates how $R_{ct}$ was changing, showing that little change was occurring at the interface over time while applying $100\,\mu A\,cm^{-2}$. $\chi_M$ for the 1 m electrolyte was calculated as $1.41 \pm 0.33$, which implied a higher "effective concentration" than the molarity that is used.[38] This is a result of depleting free solvent, and increasing bound solvent, as is illustrated by the increasing $868\,cm^{-1}$ peak intensity from Fig. 2b.

The Nyquist plot from PEIS, which was run after 4 h rest, is plotted in Fig. 3e. The ionic conductivity ($\kappa$) was determined with Eq. (5).

$$\kappa = \frac{L}{R_{bulk} A} \quad (5)$$

Where $A$ is the electrode area. Using an optical cell makes it especially straightforward and accurate to measure $L$, which can be a problem when conventionally measuring $D_{app}$ and $\kappa$, where $L$ can be difficult to control. For 1 m, $\kappa = 2.70 \pm 0.03\,mS\,cm^{-1}$ and

$R_{ct} = 92 \pm 12\,\Omega\,cm^2$. $R_{ct}$ was assumed to be a combination of both resistances of classical charge transfer, and SEI resistance.[39] For further information on error estimation calculations, refer to Supplementary Note 2.

**Effect of concentration**. The above method was applied to 0.25 m, 0.5 m, 1 m, 1.5 m and 2 m electrolyte solutions to compare $\kappa$, $R_{ct}$, $D_{app}$, $t_+^0$ and $\chi_M$, for a broader understanding of the electrolyte system. Repeat runs were taken at each concentration, both with and without PEIS between each line scan to check the PEIS measurement was not affecting the gradient measurement.

**Ionic conductivity and resistance of charge transfer**. Supplementary Fig. 4 indicates clear differences in impedance with concentration. $\kappa$ vs. concentration was plotted (Fig. 4a) and fitted with the function proposed by Casteel and Amis,[40] see Supplementary Eq. (S22). $\kappa_{max}$ was calculated as $2.67\,mS\,cm^{-1}$ at 0.99 M. $\kappa_{max}$ is lower than more conventional electrolytes, such as LP30 ($\sim11\,mS\,cm^{-1}$), due to its increased viscosity. $c_{max}$ is very similar to most binary non-aqueous electrolytes based on carbonates or ethers, i.e., 1 M.[6]

As noted in Fig. 4b, there was an evident decrease in $R_{ct}$ going from 0.25 m to 1 m, but a flattening as one moved to higher concentrations. The decrease in $R_{ct}$ at lower concentrations is due to the dependence of concentration on interfacial kinetics, as is described by the exchange current density in electrochemical kinetics.[37] The plateauing at higher concentrations could be a result of an increased resistivity of the SEI or an increased charge-transfer resistance from an increase in solvation energy. Structurally, an increased concentration leads to the formation of more CIP and AGG-type structures (see Fig. 2b) which inform the SEI composition more than at lower concentrations. Furthermore, the change in solvation structure may lead to

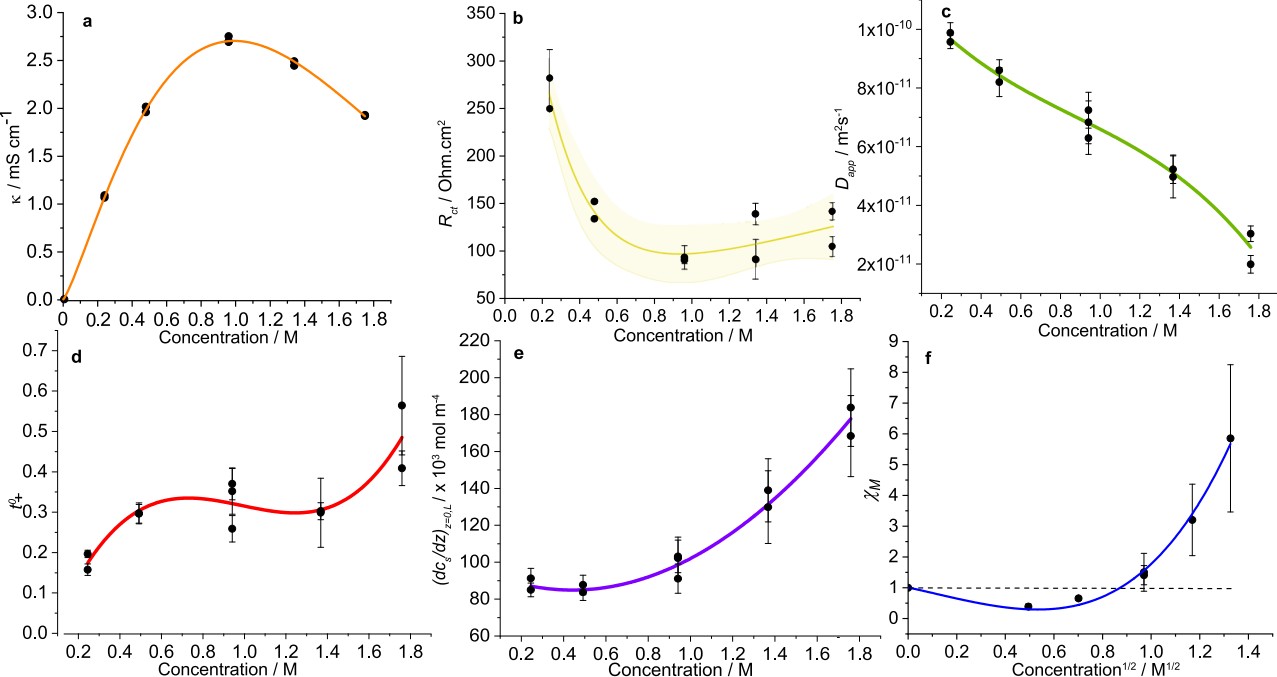

**Fig. 4 LiFSI in G4 concentration-dependent transport and thermodynamic properties. a** $\kappa$, fitted with the Casteel–Amis equation, $\kappa_{max}$ was calculated as $2.67\,mS\,cm^{-1}$ and $c_{max}$ as 0.99 M. **b** $R_{ct}$, showing an initial drop with concentration, then a stabilising at 100–125 $\Omega\,cm^2$. **c** $D_{app}$, observable was a decrease in $D_{app}$ with concentration. **d** $t_+^0$ showing an increase at low concentrations, then generally plateauing, and increasing again at 2 m. **e** $(dc/dz)_{z=0,L}$, which was quite constant at low concentrations but increased rapidly after 1 m. This was concluded to be a result of a dramatic decrease in $D_{app}$ with concentration. **f** $\chi_M$, which had values below 1 from 0 to 0.5 m, indicating association is dominating the thermodynamics and then begins to increase rapidly as solvation effects start to dominate, with ion-solvent effects out-competing ion-ion effects ($\chi_M = 1$) at $\sim0.72$ M.

an increase in activation overpotential if the solvation energy increases.

**Diffusion coefficient**. Figure 4c shows how $D_{app}$ magnitude was affected by changes in LiFSI concentration. Generally, we observed a decrease in $D_{app}$ with increasing concentration, which is common for electrolyte solutions. Grundy et al. reported the same trend with LiTFSI in G4.[41] The absolute values appeared to agree with Grundy's findings, although their chosen salt was LiTFSI, so would expect to have a slightly lower $D_{app}$. Compared to other lithium-ion electrolytes, $D_{app}$ was quite low, with 1 M PC and EMC-based electrolytes exhibiting values 2–6 times higher.[7,11] Decreasing $D_{app}$ with concentration is a result of an increasing association of ions and increasing complexation of $Li^+$ with G4; an increasing frictional interaction between ions and solvent, as described by Stefan and Maxwell.[42]

**Transference number**. Figure 4d shows how $t_+^0$ was affected by concentration. As explicitly stated by Bazak et al., to measure $t_+^0$ with accuracy, a very precise measurement of local interfacial concentration gradients is required, due to the high sensitivity of $t_+^0$ with small changes in $dc_s/dz|_{z=0,L}$. This, along with error propagation from calculating $D_{app}$, could lead to poor reliability and large error, explaining the range of error in Fig. 4d, especially at high concentrations. That said, after multiple measurements, we noticed a trend, where $t_+^0$ generally increased with concentration, although it was quite constant at moderate concentrations. One should note that if the solvent velocity factor was not included (i.e., to calculate $t_+$), it would appear the transference number has an even more pronounced increase with concentration. This underlines the importance of including this factor in transference number calculations. Within this concentration range, Grundy noticed a similar $t_+^0$ trend. A physical explanation for this shape remains contentious. We speculate that at lower concentrations the $Li^+$ ion is more solvated than $FSI^-$, which remains free. With increasing concentration, the $FSI^-$ becomes partially solvated, and so $t_+^0$ increases.

Perhaps most importantly, and so to reiterate, $dc_s/dz|_{z=0,L}$ is a function of electrolyte properties $D_{app}$ and $t_+^0$, and current density. This interfacial gradient is not easily or readily characterised but is of primary importance for understanding poor electrolyte performance in LIBs, and also in understanding the likelihood of dendrite growth in alkali metal batteries.[5] Fig. 4e shows how $dc_s/dz|_{z=0,L}$ changed with concentration. Although fairly constant at lower concentrations, it increased very dramatically at higher concentrations. With a constant current applied to all concentrations, this was only due to changes in $D_{app}$ and $t_+^0$, which have been characterised. With a fivefold drop in $D_{app}$ over the concentration range, the large increase in $dc_s/dz|_{z=0,L}$ at high concentrations was primarily due to the drop in $D_{app}$.

**Thermodynamic factor**. Illustrated in Fig. 4f, the electrolyte behaved according to extended Debye–Huckel theory described in Supplementary Note 7; $\chi_M$ is plotted against the square root of concentration.[38] Similar to Hou[7] and Wang's[11] approach, the data were fitted with an extended Debye–Huckel Guggenheim equation; an $x^{1/2}$ power series.[43] This shape is akin to those of other organic-based electrolytes;[7,11,13,14] at low concentrations, a drop in $\chi_M$ suggests long-range ion-ion association dominates, but at higher concentrations, it is shorter range solvation effects that dominate. With increasing concentration, the solvent becomes increasingly bound, which decreases its vapour pressure, leading to an increase in salt activity and therefore $\chi_M$. The extent to whether association dominates $\chi_M$ values is primarily influenced

by solvent physicochemical properties, like relative dielectric constant ($\epsilon_r$). For instance, a high $\epsilon_r$ will allow for more salt dissociation, as it stabilises isolated ions, and therefore the effects of association do not appear. Wang et al. showed this by comparing a PC-based electrolyte with an EMC-based electrolyte.[11] G4 has a $\epsilon_r$ that lies between PC and EMC (7.91 vs. 64.92 and 2.96, respectively), and its $\chi_M$ dependence on concentration agrees with this. The concentration at which the curve crosses $\chi_M = 1$ is indicative of when ion-solvent effects out-compete ion-ion effects. For EMC and PC, it was 1.01 M and 0.12 M respectively, and for this system, it was ~0.72 M.

Using Raman, the extent of associated species forming is shown in Fig. 2c: the ~ 717 cm$^{-1}$ peak begins to shift, and shoulder peaks start to emerge, demonstrating increased $Li^+$–$FSI^-$ interaction. Also, Bockris suggests triple-ion formation occurs when $\epsilon_r < 15$, indicating triplets and perhaps clusters are likely to form in G4.[38] Supplementary Figure 8c shows the free $FSI^-$ ion becoming a minor $FSI^-$ species at ~0.5 m as more aggregated species are dominating. As mentioned above, this evidence would suggest a decrease in $\chi_M$ with concentration.

However, the decreasing number of free solvent molecules available to bind and stabilise $Li^+$, shown in Fig. 2b, leads to an increase in ion-solvent interaction and therefore an increase in $\chi_M$. The emergence and increasing intensity of the 868 cm$^{-1}$ peaks correlate with increasing $\chi_M$. As an analysis in Supplementary Note 7 shows, at concentrations 1.5-2 m, there is more bound solvent than free.

In summary, the introduction of solvent structures like $[Li(G4)]^+$ and the formation of SSIPs, CIPs and AGGs have competing effects on the observed $\chi_M$ value. The fairly low G4 $\epsilon_r$ leads to SSIPs and CIPs at low concentrations resulting in a decrease in $\chi_M$; with increasing concentration more $[Li(G4)]^+$-like solvent structures form, resulting in an increase in $\chi_M$.

By measuring parameters $\kappa$, $D_{app}$, $t_+^0$ and $\chi_M$ one can implement the values into a Doyle–Fuller–Newman (DFN) model and simulate symmetric and full cell performance. We performed these simulations using the Batteries and Fuel Cells Module in COMSOL Multiphysics 5.5. Firstly, a 1 m LiFSI in G4 concentration gradient in a symmetric cell was modelled at 15 μm, 50 μm and 100 μm, at 5 mA cm$^{-2}$. Then, we simulated a LIB cell using LiFSI in G4 with a thickness of 20 μm to understand how electrolyte concentration and properties, and C-rate affects LIB cell performance. It was evident that using 1 m LiFSI in G4 led to the lowest overpotential in the LIB simulation and the highest final capacity at a 3 V cut-off. For details on this study, refer to Supplementary Discussion 1.

**Stefan–Maxwell diffusion coefficients**. For a more in-depth understanding of the diffusional behaviour of each individual species and their interactions with the other species in solution, the Stefan–Maxwell coefficients were calculated using concentrated-solution theory, developed by Newman and Thomas-Alyea.[21] Stefan–Maxwell diffusion theory provides information on ion-solvent and ion-ion frictional interactions, which is an important consideration when describing diffusion in multicomponent systems. Using Supplementary Eqs. (S28)–(32), $\mathfrak{D}$ and the other Stefan–Maxwell coefficients, $\mathfrak{D}_{0+}$, $\mathfrak{D}_{0-}$ and $\mathfrak{D}_{+-}$ are plotted. $\mathfrak{D}$ is the thermodynamic diffusion coefficient, which is affected by salt chemical potential gradients rather than concentration gradients; $\mathfrak{D}_{0+}$ and $\mathfrak{D}_{0-}$ are the Stefan–Maxwell intermolecular diffusion coefficients of the cation (+) and anion (−) respectively, and their interaction with the solvent; and $\mathfrak{D}_{+-}$ is the coefficient that describes the interaction between cation and anion.

$\mathfrak{D}$ followed an exponential decay with concentration, see Fig. 5a. $\mathfrak{D}_{0+}$ and $\mathfrak{D}_{0-}$ were very large at low concentrations, with

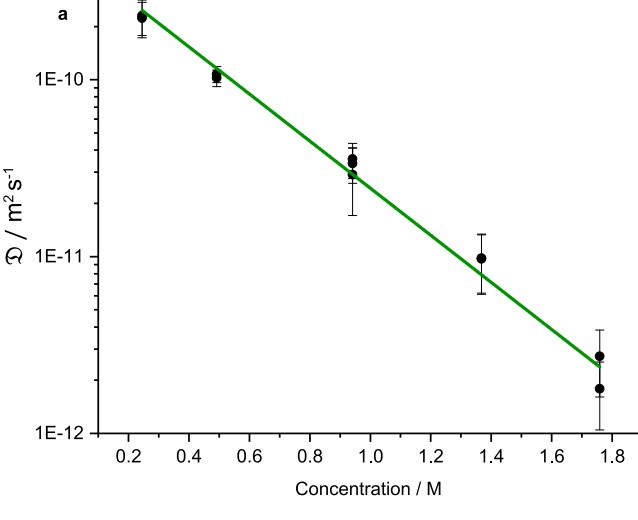

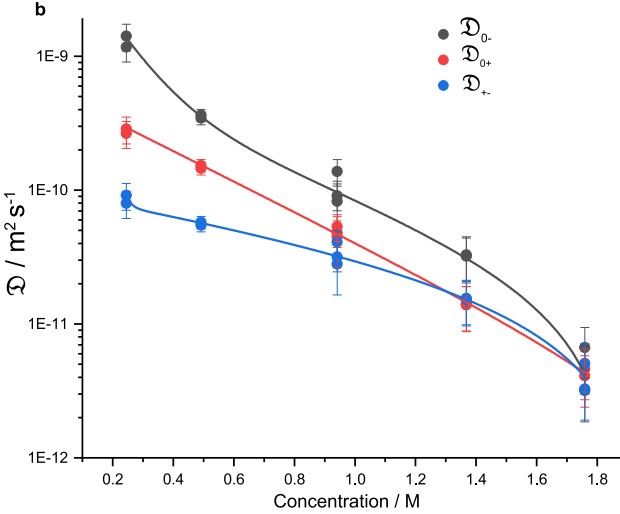

**Fig. 5 Stefan–Maxwell analysis. a** Exponential decay of LiFSI thermodynamic diffusion coefficient ($\mathfrak{D}$) with concentration. **b** Stefan–Maxwell coefficients $\mathfrak{D}_{0+}$, $\mathfrak{D}_{0-}$ and $\mathfrak{D}_{+-}$, all decreasing with concentration. $\mathfrak{D}_{0-}$ was generally the highest value across concentrations, indicating FSI$^-$ had the least frictional interaction with G4, while FSI$^-$'s frictional interaction Li$^+$ was high. At 2 m, each Stefan–Maxwell coefficient was very similar, within error, which could illustrate a change in transport mechanism, from vehicular to ion-hopping. The error bars in (**a**) and (**b**) represent the standard error and propagation from previous calculations, see Supplementary Note 8.

$\mathfrak{D}_{0-}$ generally higher than $\mathfrak{D}_{0+}$. This is a result of increased movement of FSI$^-$ due to weaker interaction with the solvent compared to Li$^+$, which is known to bind to G4 strongly, as evidenced by $t_+^0 < 0.5$. The general decreasing trend of both $\mathfrak{D}_{0+}$ and $\mathfrak{D}_{0-}$ is due to the increasing drag on the solvated complexes. $\mathfrak{D}_{+-}$ was the lowest of each of the Stefan–Maxwell coefficients at low concentrations because of the poor movement of paired species. Interestingly at 2 m, each of the coefficients was of similar magnitude, within error. As free solvent is depleted, aggregates and/or even domains of ions begin to form, and we speculate the mechanism of transport begins to transition from vehicular movement to ion-hopping. Imide-based salts in G4 are sometimes described as "solvate" ionic liquids at high concentrations.[36,44] The strong [Li(G4)]$^+$ complexes forming and their weak interaction with FSI$^-$ the solution could be

described as a "quasi" ionic liquid. As solvent molecules begin to deplete in solution, this description becomes apt.

## Discussion

A method for the full characterisation of electrolyte systems has been presented using operando Raman microspectroscopy in tandem with PEIS. The Fickian diffusion coefficient, transference number, thermodynamic factor, ionic conductivity, the resistance of charge-transfer and Stefan–Maxwell coefficients of LiFSI in G4 have been studied through the formation and analysis of concentration gradients. An understanding of how each listed property is affected by bulk concentration was described, with structural information from Raman also providing insight into how solvent and electrolyte structure affects electrolyte properties. Specifically, we noted the interfacial concentration gradient increased with increasing bulk concentration, which was primarily due to decreasing $D_{app}$. So, although much attention is placed on increasing $t_+^0$ to reduce $\eta_c$, focusing on increasing $D_{app}$ could be a more valuable pursuit for future work. Also, this was the first time $\chi_M$ was measured using concentration gradient visualisation; we hope our description will be a useful tool for more frequent $\chi_M$ characterisation. The full significance of interfacial concentration gradients and $\chi_M$ on LMB performance and its influence on lithium plating morphology could be compelling further work.

For explicit context, the measurement of $\kappa$, $R_{ct}$, $D_{app}$, $t_+^0$ and $\chi_M$ provides a full description of the electrolyte, which can be used in the theoretical modelling of battery systems, and also provides an explanation of potential shortcomings of measured electrolytes in Li-ion and "beyond Li-ion" systems. We showed that at discharge rates of 1C and 4C in our full-cell simulations, the 1m electrolyte exhibited the least overpotential and attained the highest average SOC.

Indeed, to successfully measure concentration gradients with Raman microspectroscopy, an isolatable and concentration-sensitive Raman band is required. Furthermore, if the required Raman band(s) are present, our described method could be extended to electrolyte systems with multiple ions. Evidence from the few other studies that have used Raman spectroscopy to monitor electrolyte concentration differences shows that many commonly used salts (e.g., LiClO$_4$[27], LiBOB[28], LiTFSI[29]) and solvents (e.g., DMC[27] and G4[28]) can successfully be studied via the presented method. The most popular salt in LIBs, LiPF$_6$, has been studied using Raman for structural analysis, although, to the best of our knowledge, not been used to test concentration changes in the solution. Although LiPF$_6$ has been shown to induce a fluorescence background in Raman spectroscopy experiments, its strong PF$_6^-$ Raman-active band would make it an ideal candidate for further studies.[45] There is little doubt, therefore, that many current and future electrolyte mixtures benefit from measuring electrolyte properties in this manner. To increase the viability of the presented method further, more advanced Raman techniques such as stimulated Raman spectroscopy (SRS) could be employed. Its utilisation would increase the sensitivity of the method and reduce the acquisition time per spectrum.[46]

Overall, we hope this work will provide an alternative method for electrolyte characterisation and a tool for progressing our communal understanding of how these properties affect Li-ion battery performance and Li$^+$ electrodeposition morphology.

## Methods

**Electrolyte description**. Lithium bis(fluorosulfonyl)imide (LiFSI) (Battery Grade —99%) was purchased from Fluorochem Ltd. Tetraethylene glycol dimethyl ether (G4) (anhydrous, 99%+) was purchased from Sigma Aldrich. Handling of LiFSI and G4 was always performed in an argon-filled glovebox (MBraun) with low H$_2$O

content (<1 ppm) and low $O_2$ content (<1 ppm). LiFSI was dried further under a high vacuum at 70 °C for 48 h. G4 has dried over 3 Å molecular sieves, which were washed and then dried for one week. All glassware was dried at 80 °C under vacuum before being used and brought into the glovebox. The $H_2O$ content of the electrolyte solutions was determined by Karl Fischer titration, also performed in an argon-filled glovebox, and recorded to be below 15 ppm of $H_2O$. Experimental information describing density measurements and partial molar volume calculations are available in Supplementary Method 1 and 2.

**Concentration gradient calibration with Raman spectroscopy**. A Renishaw inVia Reflex laser confocal Raman microscope equipped with a near-IR 785 nm laser, a 5× magnification objective (Leica, 0.12 NA, 14 mm WD), providing a 4.8 μm spot size, and a 90° mirror was used to collect Raman spectra of the prepared calibration solutions. Specifically, we recorded spectra with a centre at 1200 $cm^{-1}$, at 5% laser power, with a 1 s exposure time for each calibration sample. Measurements were repeated 20 times. Following the acquisition, we removed the background, using the Renishaw WiRE 5.5 software, and normalised all spectra against the intensity of the 1471 $cm^{-1}$ band. The latter representing the $CH_2$ bending/scissoring mode of G4. We further used this feature to normalise the line scan spectra. The Raman band at 717 $cm^{-1}$, corresponding to S–N–S vibrations of the FSI$^-$ anion, was finally integrated to produce the calibration curve shown in Fig. 2c of the main text.

**Cell construction**. The operando cell was constructed in an Ar-filled glovebox. First, two stainless steel pistons were designed to fit inside a quartz glass tube 20 mm in length, with an inner diameter (ID) of 8 mm and an outer diameter (OD) of 10 mm. An interelectrode distance of 15 mm was chosen because it was sufficiently large to detect a concentration gradient with good spectral resolution and the diffusion layer not to progress too quickly into the centre of the cell, which would make the fitting equation invalid. If it were longer the diffusion layer would not progress quickly enough, and the measurement would take substantially longer than 36 h. Each piston was further equipped with an O-ring made of FFKM for increased chemical resistance against ethers. The lithium metal foil (99.9%, 750 μm thickness, Alfa-Aesar) acting as electrodes for the symmetric cell, was prepared by scraping off the native oxide layer of the foils and calendaring to 300 μm in thickness. From here, circular lithium discs were prepared with a diameter of 8 mm and placed onto each of the stainless steel pistons. One piston was then placed inside the quartz tube, ~1 mL of electrolyte was added, and then the second piston was introduced on the opposite side cell, sealing the cell. Care was taken not to introduce any gas bubbles into the system. Before any tests were run, the cell was set up inside the Raman microscope and connected to a Biologic SP150 potentiostat (Supplementary Fig. 2). Important to note, the cell was vertically placed on the sample stage to avoid natural convection (Supplementary Fig. 3a).

**PEIS**. Before any line scan or current was applied, PEIS was performed on the cell. First, a PEIS scan with voltage amplitude ($V_a$) 100 mV and frequencies from 100 kHz to 1 Hz, was performed every hour, to monitor the stability of the interface. The large voltage amplitude was chosen for a reasonable signal-to-noise ratio. Linearity was still maintained. The ionic conductivity ($\kappa$), from $R_{bulk}$ and resistance of charge transfer ($R_{ct}$) were collected after 4 h. The data were then fitted using Biologic EC-lab V11.26 software with the equivalent circuit $(Q_{bulk}/R_{bulk})+(Q_{dl}/R_{ct})$, with $R_{bulk}$ and $R_{ct}$ representing the bulk resistance of the electrolyte and charge transfer respectively, and $Q_{bulk}$ and $Q_{dl}$ are the constant phase element of electrolyte and double layer respectively.

**CP and operando Raman line scan measurements**. A constant current was applied to the operando cell for 24–48 h, allowing a concentration gradient to form. We chose 100 μA $cm^{-2}$ as the operating current to avoid dendrite formation, and also to allow a concentration gradient of sufficient magnitude to form. Both visually, and there is no drop in $R_{ct}$ with time, provided evidence that there was no dendrite growth at 100 μA $cm^{-2}$. Supplementary Fig. 3c shows how the current density affected the concentration gradient. Supplementary Fig. 3d shows that mossy dendritic structures formed at 400 μA $cm^{-2}$. In Supplementary Fig. 3c, the gradient that formed at this current density was very steep, with dendrites blocking the signal close to the plating electrode.

As the current was applied, a 1D line scan in the $z$-direction was started, with the same laser settings as the calibration curve. A laser power of 5% was deemed optimal as it provided good spectral resolution, whilst showing no evidence of thermal heating or oxidation of the solvent. This is illustrated in Supplementary Fig. 3e, which shows no change in the solvent peak with time, and therefore no evidence of decomposition products. Moreover, there was no emergence of additional peaks in the spectra after repeated line scans. Also, there was no decrease in the overpotential with time, indicating no thermal heating. For the starting point of each line scan, we focused as close to the electrode surface as possible, at a defined depth. Confocal microscopy uses a pinhole aperture to block out-of-focus light, measuring spectra in a precise plane of focus. This led us to gather spectra at defined depths (the centre point of the cell) with accurate spot size. The gradient evolution during a line scan acquisition, under the selected operating conditions of the cell, was determined to be within the signal-to-noise error for the acquired

Raman spectra. For quantifying $\chi_M$, a PEIS measurement was performed before each line scan acquisition. These measurements were performed with a $V_a = 100$ mV and frequency range of 100 kHz to 1 Hz, resulting in each run taking ~30 s. To make sure that these PEIS measurements and the associated CP interruption did not negatively affect the $D_{app}$ and $t_+^0$ measurements determined with the Raman line scans, a second set of experiments was performed without PEIS interruption. No visible difference between these experiments could be determined. Using Renishaw's WiRE5.5 software, each spectra had its background removed, then using a Python script each spectra was compared to the calibration curve to extract the electrolyte concentration at each point along with each line scan.

## Data availability

The authors declare that all data supporting the findings of this study are included within the paper and its Supplementary Information. Source data are available from the corresponding author upon reasonable request.

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

## Acknowledgements
The authors would like to acknowledge the ISCF Faraday Challenge projects SOLBAT [grant number FIRG007] and LiSTAR [grant number FIRG014] as well as the Henry Royce Institute (through UK Engineering and Physical Science Research Council grant EP/R010145/1) for capital equipment. Also, we would like to thank Andrew A. Wang for his assistance with the density measurements, and many fruitful discussions; Shobhan Dhir for insightful conversations and constructive comments; Giulia Galatolo for her Fig. 1 design, and Claire Halloran for some great earlier concentration gradient analysis work.

## Author contributions
J.F. and M.P. conceived the idea and designed the experiments. J.F. performed all the experiments, measurements, simulations, analysis and wrote the manuscript with input from all authors. J.I. helped with the initial operando Raman microspectroscopy experiments and provided clarity and helpful discussion on the interpretation of spectroscopic data. F.L.M. provided expertise relating to the electrochemical data. M.P. supervised the design of the project and provided frequent input in the interpretation of all results.

## Competing interests
The authors declare no competing interests.
