## [Peer Review File · Nature Communications]

Reviewer #1 (Remarks to the Author):

The presented paper introduces a new experimental methodology to determine fundamental physicochemical electrolyte properties associated with ion transport in batteries. The study is well conceived, planned, executed, and summarized judging from the quality of the data, its analysis, conclusions, and the level of writing. The level of technical details is appropriate and the description of the setup including photos and general experimental considerations is much appreciated. The publication of such rigorous work in "high impact" journals, such as Nature Communications, is highly appreciated considering the generally high noise level of scientific output at that impact-level. The presented work will be of significant use for the research community concerned with electrolytes in batteries and related applications. I therefore recommend accepting the manuscript for publication without further revision.

Reviewer #2 (Remarks to the Author):

This manuscript reports an alternative method for electrolyte characterization by measuring the electrolyte concentration gradient over time using operando Raman micro spectroscopy, in conjunction with impedance spectroscopy and theoretical simulations. The manuscript explains how these properties affect Li-ion battery performance and Li⁺ electrodeposition morphology. The work is new and convincing, and of interest to the battery community. The manuscript is well written and can be accepted with some of the minor revisions noted below.

I. Please explain in detail the reasons for selecting lithium bis(fluorosulfonyl)imide (LiFSI) in tetraglyme (G4) as a model system and how this particular electrolyte salt-solvent combination forms a more desirable solid electrolyte interphase (SEI) on lithium metal compared to conventional lithium ion battery electrolytes.

II. In Fig. 2 (c), what is the error for the data points in these measurements? Better to show the error bar in the graph.

III. On page 4, the authors made a statement, "To ensure the validity of the diffusion, an interelectrode distance of 1.5 cm was chosen". Describe the reasons for choosing an interelectrode spacing of 1.5 cm in these experiments.

IV. In Fig. 3 (d), Why do the errors for the ηc function increase significantly as we go to higher concentration ratios?

V. In Figs. 4 (a, b) & 5 (a, b), How large is the error for the data points?

VI. The authors simulated a full LIB cell to understand the effects of electrolyte concentration and C-rate on LIB cell performance. The simulation of the discharge curve at higher currents (4C) shows that the cell with the concentration of 0.25m and 0.5m performs poorly compared to 1C. What are the reasons for this behavior of discharge curves?

VII. It is better to compare the deviation between simulated and measured voltage curves at higher discharge current rates for these concentrations (0.25-2 m)?

VIII. Required to strengthen the conclusions

Reviewer #3 (Remarks to the Author):

This paper details a method to determine some of the key parameters describing electrolytes including diffusion coefficient, concentration gradient, conductivity, and charge transfer resistance that relies on Raman spectroscopy to give additional information on concentration gradients. They then use the Ficks Diffusion laws. Transference number in particular is very difficult to measure, so this is a useful contribution to the field. Some comments below.

1) One of the key premise upon which this study is built is that that the 717 cm^{-1} Raman band corresponding to the FSI- anion changes directly proportional to the concentration. They use a calibration curve using electrolytes with different molarities. Clearly this assumption is valid under static conditions based on the data presented. However, what is less clear to me is how valid this is under dynamic conditions. As a bias is applied, Li will move one way and the anion the other. It seems to me that it is very possible that the correlation between the Li concentration and the FSI ions will decouple under these conditions. While the general trend the observe with more the FSI anions concentrating on the positive electrode and depleting next to the negative electrode is undoubtedly accurate, the magnitude of the numbers seems less certain under dynamic conditions to me. More justification is needed for the use of the static calibration under dynamic conditions.

2) One of the biggest discrepancies between this work and real cells is that in this work the electrodes are spaced 1.3 cm apart. This makes it very easy to measure a line scan in Raman between them. A real cell however has a separator and electrolyte thickness of $\sim 20\mu\text{m}$. How valid is the present work to a practical cell dimensions? Would the numbers measured differ at all for a thin electrolyte separation. Perhaps some COMSOL modeling could back up and justify the generalization here.

3) While the work is of interest, perhaps its biggest limitation is only one electrolyte is presented here, and that electrolyte is not a common electrolyte for liquid batteries. I would very much like to see at least proof of concept using this Raman/potentiostat measurement for commercial electrolytes. If the LiPF₆ is not doable based on the Raman spectra, then I would like to see if for the same LiFSI, but in a more likely solvent to be found in a practical cell.

4) For the calibration curve with concentration, this seems mostly depnedable, however the refractive index of a solution will also be dependent on the salt concentration and will affect the Raman intensity (see Journal of the Electrochemical Society, 166(2), p.A178). Can the authors comment on how the disentangled the two effects?

Characterising lithium-ion electrolytes via operando Raman microspectroscopy

Response to reviewers' comments

Jack Fawdon^a, Johannes Ihli^{a,b}, Fabio La Mantia^c, Mauro Pasta^{*a}

^aDepartment of Materials, University of Oxford

*E- mail: mauro.pasta@materials.ox.ac.uk

^bPaul Scherrer Institut

^cUniversität Bremen, Energiespeicher-und Energiewandlersysteme

We'd like to thank all the Reviewers for their constructive and insightful comments. Changes implemented to the original document are reported in blue and highlighted in yellow for the direct quote to the revised manuscript.

Reviewer #1 (Remarks to the Author):

The presented paper introduces a new experimental methodology to determine fundamental physicochemical electrolyte properties associated with ion transport in batteries. The study is well conceived, planned, executed, and summarized judging from the quality of the data, its analysis, conclusions, and the level of writing. The level of technical details is appropriate and the description of the setup including photos and general experimental considerations is much appreciated. The publication of such rigorous work in "high impact" journals, such Nature Communication, is highly appreciated considering the generally high noise level of scientific output at that impact-level. The presented work will be of significant use for the research community concerned with electrolytes in batteries and related applications. I therefore recommend accepting the manuscript for publication without further revision.

We would like to thank the Reviewer for their positive comments, noting our work's significance for the research community working with electrolytes for batteries and other applications.

Reviewer #2 (Remarks to the Author):

This manuscript reports an alternative method for electrolyte characterization by measuring the electrolyte concentration gradient over time using operando Raman micro spectroscopy, in conjunction with impedance spectroscopy and theoretical simulations. The manuscript explains how these properties affect Li-ion battery performance and Li⁺ electrodeposition morphology.

The work is new and convincing, and of interest to the battery community. The manuscript is well written and can be accepted with some of the minor revisions noted below.

We thank the Reviewer for their positive comments, and we appreciate the opportunity to improve our manuscript by addressing the questions they have.

I. Please explain in detail the reasons for selecting lithium bis(fluorosulfonyl)imide (LiFSI) in tetraglyme (G4) as a model system and how this particular electrolyte salt-solvent combination forms a more desirable solid electrolyte interphase (SEI) on lithium metal compared to conventional lithium ion battery electrolytes.

As a model system we wanted an electrolyte that is being proposed as a lithium metal battery candidate; an anion with a clear vibration active mode, and a solvent that is stable against lithium metal. Very few of LMB electrolyte candidates are being fully characterised, with the most rigorous studies opting for LiPF₆ in a carbonate-based solvent -- a common LIB electrolyte system. We were therefore presenting a methodology for fully characterising electrolytes in general while also elucidating information on an LMB electrolyte candidate that is so rarely fully characterised.

Specifically, as we mentioned in the text, LiFSI is an increasingly popular salt for LMBs:

“LiFSI is an increasingly popular salt used in lithium metal anode studies due to its low viscosity in typical solvents and desirable decomposition products that form the solid electrolyte interphase (SEI).”

When using LiFSI, it's been shown the SEI formed on lithium metal is “fluoride-rich”, especially at high concentrations due it being primarily anion-derived. The surface LiF has a very low electronic conductivity, which has been proposed as a reason for reduced dendrite formation when using LiFSI.

Glymes, of which G4 is included, have been proposed as a stable solvent for LMB electrolytes due to their high cathodic stability and low viscosity. As we mention in the text:

“Glymes are also popular for use against lithium metal (in LMBs) because of their cathodic stability and low viscosity. They have been studied for their use in Li-O₂ and Li-S batteries in particular.”

Additionally, tetraglyme (G4) based electrolytes exhibit low flammability, low volatility and cathodic stability below 0V vs. Li⁺/Li. Due to its high cathodic stability, it is less likely to contribute to SEI formation, leaving the lithium surface pristine for stable electrodeposition. Carbonates, by contrast, are less stable against lithium, and have shown to lead to less stable electrodeposition.

Added to the manuscript, Introduction, Page 3:

“LiFSI is an increasingly popular salt used in lithium metal anode studies due to its low viscosity in typical solvents and desirable decomposition products (e.g. LiF) that form the solid electrolyte interphase (SEI). Fluoride-rich SEI layers have shown to be effective at suppressing dendrite growth due to the electronic insulation and high surface energy of LiF. [34, 35]. Glymes are also popular for use against lithium metal (in LMBs) because of their cathodic stability and low viscosity. G4 has been shown to exhibit low flammability, low volatility and cathodic stabilities below 0V vs. Li⁺/Li. [36] They have been studied for their use in Li-O₂ and Li-S batteries in particular.”

[34]: Ko, J. & Yoon, Y. S. Recent progress in LiF materials for safe lithium metal anode of rechargeable batteries: Is LiF the key to commercializing Li metal batteries? *Ceramics International* **45**, 30–49 (2019)

[35]: Liu, Z. et al. Interfacial Study on Solid Electrolyte Interphase at Li Metal Anode: Implication For Li Dendrite Growth *Journal of The Electrochemical Society* **163**, A592–A598 (2016)

[36]: Terada, S., et al. Liquid structures and transport properties of lithium bis(fluorosulfonyl)amide/glyme solvate ionic liquids for lithium batteries. *Australian Journal of Chemistry* **72**, 70–80 (2019)

II. In Fig. 2 (c), what is the error for the data points in these measurements? Better to show the error bar in the graph.

We thank the reviewer for their suggestion.

We have added the error in the x- and y-direction for Figure 2c, where in the x-direction, the error is based on the propagation of uncertainties related to balance error when weighing salt and solvent, and the error from the density measurement to calculate the molarity of each solution. To calculate the error in the y-direction, we ran three Raman measurements (same settings as the line scan: static scan, 785nm, laser 20 accumulations, 5% power) at various concentrations and noted a concentration independent variation in FSI peak area after normalisation. Each concentration showed a ~3% standard deviation, which is shown in the error bar.

Figure 2c is amended, page 3:

Figure 2: Intermolecular structure of lithium bis(fluorosulfonyl)imide (LiFSI) in tetraglyme (G4), and correlation of 717 cm⁻¹ FSI peak area with Li⁺, as determined by Raman microspectroscopy. a) Raman spectrum of 2m LiFSI in G4 normalised to peak intensity at 1471 cm⁻¹; identifying the 717 cm⁻¹ FSI S-N-S bending mode peak, 868 cm⁻¹ [Li(G4)]⁺ crown ether-like breathing mode peak, 1471 cm⁻¹ -CH₂ bending/scissoring mode peak. b) Normalised 717cm⁻¹ peak, showing broadening of the peak with concentration, indicating the evolution of CIPs and AGGs, as illustrated. c) Calibration curve equating normalised 717 cm⁻¹ peak area to concentration, which was used to determine the concentration gradient via operando Raman microspectroscopy 1D line scans.

III. On page 4, the authors made a statement, “To ensure the validity of the diffusion, an interelectrode distance of 1.5 cm was chosen”. Describe the reasons for choosing an interelectrode spacing of 1.5 cm in these experiments.

Primarily, we required the concentration gradient to be large enough for us to detect with our Raman microspectroscopy setup; the larger the interelectrode distance, the larger the evolving concentration gradient will become. Our experiments showed that 1.5 cm was sufficiently large to detect the concentration gradient, with good spectral resolution. If it were larger, it would take substantially longer than 36 h for us to measure the progression of the diffusion layer.

Also, as we note in the main text Methods “Cell Construction” section (Page 10), if the interelectrode distance was smaller, the diffusion layer would progress too quickly into the centre of the cell during repeated line scans, rendering the fitting equation invalid.

Added to the main text Methods “Cell Construction” section (Page 10):

“An interelectrode distance of 15 mm was chosen because it was sufficiently large to detect a concentration gradient with good spectral resolution and also large enough for the diffusion layer not to progress too quickly into the centre of the cell, which would make the fitting equation invalid. If it were larger the diffusion layer would not progress quickly enough, and the measurement would take substantially longer than 36 h.”

IV. In Fig. 3 (d), Why do the errors for the η_c function increase significantly as we go to higher concentration ratios?

Due to error propagation, the error bar here is derived from the transference number error. This error was calculated using:

$$\sigma_{\chi_M} = \frac{\sigma_{t_+}}{t_+^0} \chi_M$$

Where σ_{χ_M} is the standard error of χ_M and σ_{t_+} is the standard error of t_+^0 .

Because χ_M is increasing, and the σ_{t_+} remains constant, σ_{χ_M} increases. However, this induces a constant percentage error of χ_M (i.e. error is remaining constant as a proportion of χ_M).

After reviewing these comments, and introducing further rigor, we added the error in the x-axis of Figure 3d showing the uncertainty in concentration at $x = 0$ and $x = L$. The associated error in the linear fit, and therefore the calculation of χ_M error, has been amended for Figure 4f.

Amended in manuscript, Figure 4d:

Figure 4d: Model system 1m LiFSI in G4, data analysis. a) The Li^+ concentration profile in 1m LiFSI in G4 electrolyte after applying $100\mu\text{A cm}^{-2}$ of current for 12h, with fitting equation 1. b)

Progression of concentration gradient in 1 m electrolyte, after 4h, 12h, 20h, 28h. c) Movement of Li^+ diffusion length L_d with $\text{time}^{1/2}$, showing linear behaviour for the D_{app} calculation. $D_{app} = 7.22 \times 10^{-11} \text{m}^2 \text{s}^{-1}$. d) Plotting equation 4, illustrating how the η_c function changes linearly with respect to the natural log of concentration ratio of each cell extreme, to measure χ_M . $\chi_M = 1.73$. e) PEIS Nyquist plot, with equivalent circuit, of 1m electrolyte, prior to current being passed, after 4h rest, equating to $\kappa = 2.70 \pm 0.03 \text{mS cm}^{-1}$ and $R_{ct} = 92 \pm 12 \Omega \cdot \text{cm}^2$.

Added to the manuscript, Results, Page 6:

“From the fitted profiles, the concentration at each interface was measured and using equation 4 below χ_M was calculated from the gradient of a weighted linear fit, see Figure 3d. The error bars represent uncertainty in interfacial concentration from the fitted profiles in the x-axis and error in the t_+^0 calculation in the y-axis.”

Amended Figure S7d, in Supplementary Information, Page 11, with the x-axis error bars for each concentration:

Supplementary Figure 7: a) Voltage of operando cell with 1m electrolyte, while passing $100 \mu\text{A cm}^{-2}$. The increase in voltage is due to the increasing magnitude of the electrolyte concentration gradient. b) $R_{bulk} + R_{ct}$ over time, showing a stable interface with no side reactions c) Concentration at cell plating and stripping electrodes using 1m electrolyte, while passing $100 \mu\text{A}$

cm^{-2} . d) Plotting equation 4 (main text) of each electrolyte concentration, illustrating each gradient and ultimately how χ_M is varying with concentration.

V. In Figs. 4 (a, b) & 5 (a, b), How large is the error for the data points?

We thank the Reviewer for their question. We have added the error to each of the requested figures and added them to the manuscript.

Added to the manuscript, Figure 4a and 4b, Results, Page 7:

Figure 4: LiFSI in G4 concentration-dependent transport and thermodynamic properties. a) R_{ct} , fitted with the Casteel-Amis equation, κ_{max} was calculated as 2.67 mS cm^{-1} and c_{max} as 0.99 M . b) R_{ct} , showing an initial drop with concentration, then a stabilising at $100\text{-}125 \Omega \cdot \text{cm}^2$. c) D_{app} , observable was a decrease in D_{app} , with concentration. d) t_{+}^0 , showing an increase at low concentrations, then generally plateauing, and increasing again at 2 m . e) $dc/dx|_{x=0,L}$, which was quite constant at low concentrations but increased rapidly after 1 m . This was concluded to be a result of a dramatic decrease in D_{app} with concentration. f) χ_M , which had values below 1 from $0\text{-}0.5 \text{ m}$, indicating association is dominating the thermodynamics, and then begins to increase rapidly as solvation effects start to dominate, with ion-solvent effects out-competing ion-ion effects ($\chi_M = 1$) at $\sim 0.72 \text{ M}$.

With such accuracy on the interelectrode distance and electrode area, the κ error was calculated from the standard deviation of R_{bulk} from repeated PEIS measurements prior to cell polarisation. There was very little change in this value so the error appears very small on the graph.

We have included a zoomed in version of the conductivity fitting (Figure 5) in the Supplementary Information, illustrating the magnitude of the error bars.

Supplementary Figure 5: Peak of the κ curve, noting the magnitude of the error when measuring R_{bulk} with PEIS

The error for R_{ct} was calculated in the same way as the κ . Because R_{ct} was less stable than κ , due to the reactivity of lithium metal, the error bars are more noticeable.

Added to the main text stating error estimation in R_{ct} and κ value, Results, Page 6 and Figure 3d caption, see above (Reviewer 2, query IV):

“Where A is the electrode area. Using an optical cell makes it especially straight-forward and accurate to measure L , which can be a problem when conventionally measuring D_{app} and κ where L can be difficult to control. For 1m, $\kappa = 2.70 \pm 0.03 \text{ mS cm}^{-1}$ and $R_{ct} = 92 \pm 12 \text{ } \Omega \cdot \text{cm}^2$. R_{ct} was assumed to be a combination of both resistances of classical charge-transfer, and SEI resistance.”

For Stefan-Maxwell, the error was propagated from each of the values required to calculate the Stefan-Maxwell diffusivities (χ_M and D_{app}).

Added to the manuscript, Figure 5 with error:

Figure 5: Stefan-Maxwell analysis. a) Exponential decay of LiFSI thermodynamic diffusion coefficient D with concentration. b) Stefan-Maxwell coefficients D_{0+} , D_{0-} and D_{-+} , all decreasing with concentration. D_{0-} was generally the highest value across concentrations, indicating FSI had the least frictional interaction with G4, while FSI's frictional interaction Li^+ was high. At 2 m, each Stefan-Maxwell coefficient was very similar, within error, which could illustrate a change in transport mechanism, from vehicular to ion-hopping.

For completeness, we have included an additional section in the Supplementary Information where we explicitly describe how we calculated the error for each of the electrolyte properties (Supplementary Information, Page 6):

2.1 Error Estimation

Presented is a summary for each parameter's calculation and error estimation. For most of our calculations when propagating error we used the multiplication/division error propagation equation 20. We will note if we use a different equation when performing propagation with other functions.

$$\sigma_x = x \sqrt{\left(\frac{\sigma_p}{p}\right)^2 + \left(\frac{\sigma_q}{q}\right)^2 + \left(\frac{\sigma_r}{r}\right)^2} \quad (20)$$

D_{app} was calculated using the weighted gradient of L_d vs. $t^{1/2}$, with L_d standard error from the fitting equation 1 (main text).

t_+^0 was determined from equation 2 (main text) using the average of $dc_s/dz|_{z=0,L}$ from each fitting. t_+^0 error was propagated from the $dc_s/dz|_{z=0,L}$ standard error along with the error from the previous D_{app} calculation.

χ_M was deduced from equation 4 (main text), by plotting $\eta_c F/2RT(1 - t_+^0)$ vs. $\ln(c_{s,z=L}/c_{s,z=0})$. The y-axis error was calculated with t_+^0 and the x-axis had an error in measuring the interfacial concentration ratio. The error in $\ln(c_{s,z=L}/c_{s,z=0})$ (σ_x) was determined from the average interfacial concentration ratio ($c_{s,z=L}/c_{s,z=0}$), which was calculated from the fitting equation (equation 1-- main text), and its standard error (σ_c). This allowed the use of the error propagation equation when using natural log:

$$\sigma_x = \frac{\sigma_c}{c_{s,z=L}/c_{s,z=0}} \quad (21)$$

χ_M was subsequently calculated from linearly fitting $\eta_c F/2RT(1 - t_+^0)$ vs. $\ln(c_{s,z=L}/c_{s,z=0})$, weighted by x and y error.

κ and R_{ct} error was calculated from the standard deviation of four R_{bulk} measurements prior to cell polarisation. With the distance and area known very accurately, σ_κ is very small, and not noticeable in Figure 4a) in the main text. A zoomed version of the κ fitting is included, highlighting the small amount of error for this measurement.

For the Stefan-Maxwell diffusion coefficients (D , D_{0+} , D_{0-} , D_{-+}), error was calculated by propagating χ_M and D_{app} error."

To introduce the error estimation sections, added to the main text, page:

"Where A is the electrode area. Using an optical cell makes it especially straight-forward and accurate to measure L, which can be a problem when conventionally measuring D_{app} and κ where L can be difficult to control. For 1m, $\kappa = 2.70 \pm 0.03 \text{ mS cm}^{-1}$ and $R_{ct} = 92 \pm 12 \text{ } \Omega \cdot \text{cm}^2$. R_{ct} was assumed to be a combination of both resistances of classical charge-transfer, and SEI resistance. For further information on error estimation calculations, refer to Supplementary Information section 2.1."

VI. The authors simulated a full LIB cell to understand the effects of electrolyte concentration and C-rate on LIB cell performance. The simulation of the discharge curve at higher currents (4C) shows that the cell with the concentration of 0.25m and 0.5m performs poorly compared to 1C. What are the reasons for this behavior of discharge curves?

We thank the Reviewer for highlighting this point.

The low concentration electrolytes performed poorly because of the depleted Li^+ concentration through the cathode during discharge. Due to the low initial concentration throughout the cell and a high C-rate, causing large concentration gradients, the concentration in the cathode drops to zero in a large portion of the porous electrode. This leads to cathode not attaining much of its capacity.

We have added an additional figure to the supplementary information to illustrate this point.

Supplementary Figure 10: a) Electrolyte concentration profile at 1C with 0.5m initial concentration. b) Electrolyte concentration profile at 4C with 0.5 m initial concentration.

Added to manuscript, Supplementary Information, Page 13:

“Supplementary Figure 10a-b shows the difference in electrolyte concentration distribution for the 0.5 m electrolyte discharged at 1C and 4C respectively. Due to the low initial concentration throughout the cell and a high C-rate causing large concentration gradients, the concentration in the cathode drops to zero in much of the electrode. This leads to a low final average SOC at a 3V cut-off.”

VII. It is better to compare the deviation between simulated and measured voltage curves at higher discharge current rates for these concentrations (0.25-2 m)?

As the Reviewer suggests, comparing simulated and experimentally measured full cell performance would be of interest. However, we believe that including full-cell experimental tests would dilute the message of this study and is beyond the scope of this work.

VIII. Required to strengthen the conclusions

In the Discussion we summarised more clearly our key findings and suggested interesting avenues for further work. We removed a paragraph in the discussion to avoid repeating ourselves.

Added to manuscript, Discussion, Page 9:

“A method for the full characterisation of electrolyte systems has been presented using *operando* Raman microspectroscopy in tandem with PEIS. The Fickian diffusion coefficient, transference number, thermodynamic factor, ionic conductivity, resistance of charge-transfer and Stefan Maxwell coefficients of LiFSI in G4 have been studied through the formation and analysis of concentration gradients. An understanding of how each listed property is affected by bulk concentration was described, with structural information from Raman also providing insight into how solvent and electrolyte structure affects electrolyte properties. Specifically, we noted the interfacial concentration gradient increased with increasing bulk concentration, which was primarily due to decreasing D_{app} . So, although much attention is placed on increasing t_+^0 to reduce

η_c , focusing on increasing D_{app} could be a more valuable pursuit for future work. Also, this was the first time χ_M was measured using concentration gradient visualisation; we hope our description will be a useful tool for more frequent χ_M characterisation. Exploring the full significance of interfacial concentration gradients and χ_M on LMB performance and its influence on lithium plating morphology would be compelling future work.

For explicit context, the measurement of κ , R_{ct} , D_{app} , t_+^0 and χ_M provides a full description of the electrolyte, which can be used in the theoretical modelling of battery systems, and also provides an explanation of potential shortcomings of measured electrolytes in Li-ion and "beyond Li-ion" systems. We showed that at discharge rates of 1C and 4C in our full-cell simulations, the 1m electrolyte exhibited the least overpotential and attained the highest average SOC.

Moreover, the visualisation of concentration gradients, and measurement of the described electrolyte properties provides a tool for the understanding of non-uniform Li^+ electrodeposition in LMBs. Although Raman spectroscopy has been used to look at interfacial $[\text{Li}^+]$ depletion in the past, the advantage of our method is that we can understand the properties that causes $[\text{Li}^+]$ drop. Additionally, our method could allow us to understand how electrolyte thermodynamics influences Li plating morphology; something that has previously not been explored."

Reviewer #3 (Remarks to the Author):

This paper details a method to determine some of the key parameters describing electrolytes including diffusion coefficient, concentration gradient, conductivity, and charge transfer resistance that relies on Raman spectroscopy to give additional information on concentration gradients. They then use the Ficks Diffusion laws. Transference number in particular is very difficult to measure, so this is a useful contribution to the field. Some comments below.

We thank the Reviewer for their comments and their probing questions.

1) One of the key premise upon which this study is built is that that the 717 cm^{-1} Raman band corresponding to the FSI- anion changes directly proportional to the concentration. They use a calibration curve using electrolytes with different molarities. Clearly this assumption is valid under static conditions based on the data presented. However, what is less clear to me is how valid this is under dynamic conditions. As a bias is applied, Li will move one way and the anion the other. It seems to me that it is very possible that the correlation between the Li concentration and the FSI ions will decouple under these conditions. While the general trend the observe with more the FSI anions concentrating on the positive electrode and depleting next to the negative electrode is undoubtedly accurate, the magnitude of the numbers seems less certain under dynamic conditions to me. More justification is needed for the use of the static calibration under dynamic conditions.

Due to electroneutrality, and as expressed by Poisson's Equation, it requires an enormous amount of energy to split charges and for the cation to move one direction and the anion to move in the opposite direction:

$$\nabla^2 \phi = -\frac{F}{\epsilon} \sum (c_+ + c_-) = -1.40 \times 10^6 (c_+ - c_-) (\text{V} \cdot \mu\text{m}^{-2})$$

Where ϕ is the electrode potential, F is the Faraday constant, ϵ is the relative dielectric constant and c_+ and c_- are the concentration of cation and anion respectively. As Cheng et al. state: "An

extremely high electric field gradient in the electrolyte e.g. $5 \text{ V } \mu\text{m}^{-2}$, the difference between [cation] and [anion] is $<5 \mu\text{M}$." [28] These potentially tiny deviations from electroneutrality are negligible in our study. We therefore conclude that in a binary electrolyte, especially within the bulk of the electrolyte, in dynamic and static conditions, $[\text{FSI}^-] = [\text{Li}^+]$. Therefore, it is reasonable to use a calibration curve gathered in static conditions.

Ref 28: Cheng, Q. et al. Operando and three-dimensional visualization of anion depletion and lithium growth by stimulated Raman scattering microscopy, Nature Communications, **9**, 1–10 (2018).

2) *One of the biggest discrepancies between this work and real cells is that in this work the electrodes are spaced 1.3 cm apart. This makes it very easy to measure a line scan in Raman between them. A real cell however has a separator and electrolyte thickness of $\sim 20 \mu\text{m}$. How valid is the present work to a practical cell dimensions? Would the numbers measured differ at all for a thin electrolyte separation. Perhaps some COMSOL modeling could back up and justify the generalization here.*

We thank the Reviewer for expressing this concern regarding the practicality of the measurements.

Our study aimed to measure fundamental electrolyte properties, with each known as being independent of electrolyte thickness. However, as the Reviewer states, the experimental setup, which had electrodes spaced 1.5 cm apart, is not practical for real cells. To address this concern, we performed COMSOL modelling tests using the experimentally measured electrolyte properties, for a symmetric cell setup at 15, 50 and 100 μm , and an LIB full cell using LiFSI in G4 with an electrolyte thickness of 52 μm . These studies are presented in the Figures S10 and S11, discussed in the SI pages 13-17, and referred to in the main text on page 8. For full cell performance we used a separator thickness of 52 μm . While reviewing these comments, we updated the simulation to reduce the separator thickness to a more practical value of 20 μm as suggested by Reviewer 3.

Added to main text, Results, Page 8:

"By measuring parameters κ , D_{app} , t_+^0 and χ_M one can implement the values into a Doyle-Fuller-Newman (DFN) model and simulate symmetric and full cell performance. We performed these simulations using the Batteries and Fuel Cells Module in COMSOL Multiphysics 5.5. Firstly, a 1 m LiFSI in G4 concentration gradient in a symmetric cell was modelled at 15 μm , 50 μm and 100 μm , at 5 mA cm^{-2} . Then, we simulated a LIB cell using LiFSI in G4 with a thickness of 20 μm to understand how electrolyte concentration and properties, and C-rate affects LIB cell performance."

The updated figure, resulting from the change in model parameter, is shown below:

Supplementary Figure 9: a-c) Simulation of the 1m LiFSI in G4 electrolyte concentration gradient forming in a lithium symmetric cell at a) 15 μm b) 50 μm c) 100 μm , at 5 mA cm^{-2} . d-f) Simulation of a full LIB cell, with porous graphite and porous LMO as the negative and positive electrodes respectively, and a separator thickness of 20 μm . d) Discharge curve simulation at 1C, illustrating that the 1 m concentration performed with the least overpotential. e) Discharge curve simulation at 4C, showing cell performance with a high dependence on concentration with 0.25 m and 0.5 m performing poorly. 1 m performed at the least overpotential and highest final SOC at 3V cut-off. f) Electrolyte concentration profile at 1C.

Interestingly, the discharge curves and concentration profiles changed minimally when changing the electrolyte thickness from 52 μm to 20 μm . The trends of each concentration performed did not change, i.e. at 4C, the low concentration electrolytes performed least well because of the drop in electrolyte concentration to zero in the cathode, see Figure S10. To see a dramatic change in performance, one would also have to change the thickness of the electrodes too.

3) While the work is of interest, perhaps its biggest limitation is only one electrolyte is presented here, and that electrolyte is not a common electrolyte for liquid batteries. I would very much like to see at least proof of concept using this Raman/potentiostat measurement for commercial electrolytes. If the LiPF₆ is not doable based on the Raman spectra, then I would like to see if for the same LiFSI, but in a more likely solvent to be found in a practical cell.

We thank the Reviewer for their comment, highlighting an important point.

The goal of our study is to present a novel methodology for fully characterising a binary electrolyte system. With this goal in mind, we selected a model system that has (a) strong and isolatable Raman bands, (b) is radiation insensitive under the tested conditions, (c) is reasonably well studied, i.e. prior literature is available for comparison and evaluation purposes, and (d) is of future research interest. Additionally, like other electrolyte characterisation studies, we wanted to concentrate on one electrolyte system at various concentrations. LiFSI in G4 fitted these criteria, being one electrolyte proposed for lithium metal batteries (LMBs) use.

The desire to investigate secondary and commercially utilized systems is shared with the Reviewer but is considered to be outside of the scope for this particular methodological demonstration. We are working on a follow-up study that among others include, LiPF_6 , a popular salt for LIB applications, as the Reviewer suggests.

The characterisation of LiPF_6 containing systems with the presented method, given first results, appears feasible. The PF_6^- Raman vibrational modes are not as clear as 717 cm^{-1} S-N-S bending mode we used in our study but the 745 cm^{-1} PF_6^- mode would likely be the best salt peak and with a relevant solvent peak used to normalise the data.

Added to manuscript, Discussion, Page 10:

“Evidence from the few other studies that have used Raman spectroscopy to monitor electrolyte concentration differences shows that many commonly used salts (e.g. LiClO_4^{27} , LiBOB^{28} , LiTFSI^{29}) and solvents (e.g. DMC^{27} and G4^{28}) can successfully be studied via the presented method. The most popular salt in LIBs, LiPF_6 , has been studied using Raman for structural analysis, although, to the best of our knowledge, not been used to test concentration changes in solution. Although LiPF_6 has been shown to induce a fluorescence background in Raman spectroscopy experiments, its strong PF_6^- Raman-active band would make it an ideal candidate for future studies.[45]”

[45]: Cabo-Fernandez, L. et al. Kerr gated Raman spectroscopy of LiPF_6 salt and LiPF_6 -based organic carbonate electrolyte for Li-ion batteries. *Physical Chemistry Chemical Physics* **21**, 23833–23842 (2019).

4) *For the calibration curve with concentration, this seems mostly dependable, however the refractive index of a solution will also be dependent on the salt concentration and will affect the Raman intensity (see Journal of the Electrochemical Society, 166(2), p.A178). Can the authors comment on how the disentangled the two effects?*

We thank the Reviewer for raising this point.

The refractive index of the imaged solution will change with the concentration of the electrolyte resulting in either a decreasing or increasing Raman intensity.

This contributing factor is already taken into account in the introduced methodology by referring to a set of calibration standards. Both measurement and calibration standards are equally subject to changes in refractive index as a function of electrolyte concentration. In other words the effect “cancels out” for the intended purpose of these measurements.

Reviewer #2 (Remarks to the Author):

The revised version of the manuscript addresses most of my concerns from the previous version. So I recommend it for acceptance without any further revision.

Reviewer #3 (Remarks to the Author):

My comments and that of the other reviewers have been well addressed. I recommend acceptance in its current form.

Reviewer #2 (Remarks to the Author):

The revised version of the manuscript addresses most of my concerns from the previous version. So I recommend it for acceptance without any further revision.

Reviewer #3 (Remarks to the Author):

My comments and that of the other reviewers have been well addressed. I recommend acceptance in its current form.

We would like to thank Reviewer #2 and #3 for their positive comments and their recommendation that our study should be accepted without further revision.